# Dynamics-Aware Unsupervised Discovery of Skills

**Archit Sharma,**[*] **Shixiang Gu, Sergey Levine, Vikash Kumar, Karol Hausman**
Google Brain
{architsh,shanegu,slevine,vikashplus,karolhausman}@google.com

## Abstract

Conventionally, model-based reinforcement learning (MBRL) aims to learn a global model for the dynamics of the environment. A good model can potentially enable planning algorithms to generate a large variety of behaviors and solve diverse tasks. However, learning an accurate model for complex dynamical systems is difficult, and even then, the model might not generalize well outside the distribution of states on which it was trained. In this work, we combine model-based learning with model-free learning of primitives that make model-based planning easy. To that end, we aim to answer the question: how can we discover skills whose outcomes are easy to predict? We propose an unsupervised learning algorithm, Dynamics-Aware Discovery of Skills (DADS), which simultaneously discovers *predictable* behaviors and learns their dynamics. Our method can leverage continuous skill spaces, theoretically, allowing us to learn infinitely many behaviors even for high-dimensional state-spaces. We demonstrate that *zero-shot planning* in the learned latent space significantly outperforms standard MBRL and model-free goal-conditioned RL, can handle sparse-reward tasks, and substantially improves over prior hierarchical RL methods for unsupervised skill discovery. We have open-sourced our implementation at: https://github.com/google-research/dads

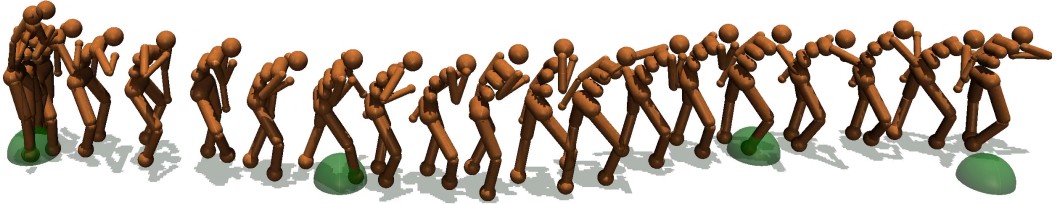

Figure 1: A humanoid agent discovers diverse locomotion primitives *without any reward* using DADS. We show zero-shot generalization to downstream tasks by composing the learned primitives using model predictive control, enabling the agent to follow an online sequence of goals (green markers) without any additional training.

## 1 Introduction

Deep reinforcement learning (RL) enables autonomous learning of diverse and complex tasks with rich sensory inputs, temporally extended goals, and challenging dynamics, such as discrete game-playing domains (Mnih et al., 2013; Silver et al., 2016), and continuous control domains including locomotion (Schulman et al., 2015; Heess et al., 2017) and manipulation (Rajeswaran et al., 2017; Kalashnikov et al., 2018; Gu et al., 2017). Most of the deep RL approaches learn a Q-function or a policy that are directly optimized for the training task, which limits their generalization to new scenarios. In contrast, MBRL methods (Li & Todorov, 2004; Deisenroth & Rasmussen, 2011; Watter et al., 2015) can acquire dynamics models that may be utilized to perform unseen tasks at test time. While this capability has been demonstrated in some of the recent works (Levine et al., 2016; Nagabandi et al., 2018; Chua et al., 2018b; Kurutach et al., 2018; Ha & Schmidhuber,

---

[*]Work done a part of the Google AI Residency program.

2018), learning an accurate global model that works for all state-action pairs can be exceedingly challenging, especially for high-dimensional system with complex and discontinuous dynamics. The problem is further exacerbated as the learned global model has limited generalization outside of the state distribution it was trained on and exploring the whole state space is generally infeasible. Can we retain the flexibility of model-based RL, while using model-free RL to acquire proficient low-level behaviors under complex dynamics?

While learning a global dynamics model that captures all the different behaviors for the entire state-space can be extremely challenging, learning a model for a specific behavior that acts only in a small part of the state-space can be much easier. For example, consider learning a model for dynamics of all gaits of a quadruped versus a model which only works for a specific gait. If we can learn many such behaviors and their corresponding dynamics, we can leverage model-predictive control to plan in the *behavior space*, as opposed to planning in the action space. The question then becomes: how do we acquire such behaviors, considering that behaviors could be random and unpredictable? To this end, we propose *Dynamics-Aware Discovery of Skills* (DADS), an unsupervised RL framework for learning low-level skills using model-free RL with the explicit aim of making model-based control easy. Skills obtained using DADS are directly optimized for *predictability*, providing a better representation on top of which predictive models can be learned. Crucially, the skills do not require any supervision to learn, and are acquired entirely through autonomous exploration. This means that the repertoire of skills and their predictive model are learned before the agent has been tasked with any goal or reward function. When a task is provided at test-time, the agent utilizes the previously learned skills and model to immediately perform the task without any further training.

The key contribution of our work is an unsupervised reinforcement learning algorithm, DADS, grounded in mutual-information-based exploration. We demonstrate that our objective can embed learned primitives in continuous spaces, which allows us to learn a large, diverse set of skills. Crucially, our algorithm also learns to model the dynamics of the skills, which enables the use of model-based planning algorithms for downstream tasks. We adapt the conventional model predictive control algorithms to plan in the space of primitives, and demonstrate that we can compose the learned primitives to solve downstream tasks without any additional training.

## 2 PRELIMINARIES

Mutual information can been used as an objective to encourage exploration in reinforcement learning (Houthooft et al., 2016; Mohamed & Rezende, 2015). According to its definition, $\mathcal{I}(X; Y) = \mathcal{H}(X) - \mathcal{H}(X \mid Y)$, maximizing mutual information $\mathcal{I}$ with respect to $Y$ amounts to maximizing the entropy $\mathcal{H}$ of $X$ while minimizing the conditional entropy $\mathcal{H}(X \mid Y)$. In the context of RL, $X$ is usually a function of the state and $Y$ a function of actions. Maximizing this objective encourages the state entropy to be high, making the underlying policy to be exploratory. Recently, multiple works (Eysenbach et al., 2018; Gregor et al., 2016; Achiam et al., 2018) apply this idea to learn diverse skills which maximally cover the state space.

To leverage planning-based control, MBRL estimates the true dynamics of the environment by learning a model $\hat{p}(s' \mid s, a)$. This allows it to predict a trajectory of states $\hat{\tau}_H = (s_t, \hat{s}_{t+1}, \ldots \hat{s}_{t+H})$ resulting from a sequence of actions without any additional interaction with the environment. While model-based RL methods have been demonstrated to be sample efficient compared to their model-free counterparts, learning an effective model for the whole state-space is challenging. An open-problem in model-based RL is to incorporate temporal abstraction in model-based control, to enable high-level planning and move-away from planning at the granular level of actions.

These seemingly unrelated ideas can be combined into a single optimization scheme, where we first discover skills (and their models) without any extrinsic reward and then compose these skills to optimize for the task defined at test time using model-based planning. At train time, we assume a Markov Decision Process (MDP) $\mathcal{M}_1 \equiv (\mathcal{S}, \mathcal{A}, p)$. The state space $\mathcal{S}$ and action space $\mathcal{A}$ are assumed to be continuous, and the $\mathcal{A}$ bounded. We assume the transition dynamics $p$ to be stochastic, such that $p : \mathcal{S} \times \mathcal{A} \times \mathcal{S} \mapsto [0, \infty)$. We learn a skill-conditioned policy $\pi(a \mid s, z)$, where the skills $z$ belongs to the space $\mathcal{Z}$, detailed in Section 3. We assume that the skills are sampled from a prior $p(z)$ over $\mathcal{Z}$. We simultaneously learn a skill-conditioned transition function $q(s' \mid s, z)$, coined as *skill-dynamics*, which predicts the transition to the next state $s'$ from the current state $s$ for the skill $z$ under the given dynamics $p$. At test time, we assume an MDP $\mathcal{M}_2 \equiv (\mathcal{S}, \mathcal{A}, p, r)$, where $\mathcal{S}, \mathcal{A}, p$

match those defined in $\mathcal{M}_1$, and the reward function $r : \mathcal{S} \times \mathcal{A} \mapsto (-\infty, \infty)$. We plan in $\mathcal{Z}$ using $q(s' \mid s, z)$ to compose the learned skills $z$ for optimizing $r$ in $\mathcal{M}_2$, which we detail in Section 4.

# 3 DYNAMICS-AWARE DISCOVERY OF SKILLS (DADS)

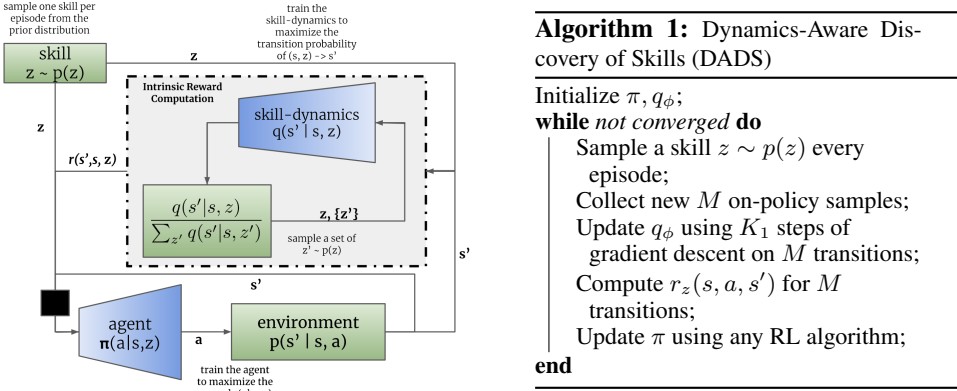

**Algorithm 1:** Dynamics-Aware Discovery of Skills (DADS)

Initialize $\pi, q_\phi$;
**while** *not converged* **do**
  Sample a skill $z \sim p(z)$ every episode;
  Collect new $M$ on-policy samples;
  Update $q_\phi$ using $K_1$ steps of gradient descent on $M$ transitions;
  Compute $r_z(s, a, s')$ for $M$ transitions;
  Update $\pi$ using any RL algorithm;
**end**

Figure 2: The agent $\pi$ interacts with the environment to produce a transition $s \to s'$. Intrinsic reward is computed by computing the transition probability under $q$ for the current skill $z$, compared to random samples from the prior $p(z)$. The agent maximizes the intrinsic reward computed for a batch of episodes, while $q$ maximizes the log-probability of the actual transitions of $(s, z) \to s'$.

We use the information theoretic paradigm of mutual information to obtain our unsupervised skill discovery algorithm. In particular, we propose to maximize the mutual information between the next state $s'$ and current skill $z$ conditioned on the current state $s$.

$$\mathcal{I}(s'; z \mid s) = \mathcal{H}(z \mid s) - \mathcal{H}(z \mid s', s) \tag{1}$$

$$= \mathcal{H}(s' \mid s) - \mathcal{H}(s' \mid s, z) \tag{2}$$

Mutual information in Equation 1 quantifies how much can be known about $s'$ given $z$ and $s$, or symmetrically, $z$ given the transition from $s \to s'$. From Equation 2, maximizing this objective corresponds to maximizing the diversity of transitions produced in the environment, that is denoted by the entropy $\mathcal{H}(s' \mid s)$, while making $z$ informative about the next state $s'$ by minimizing the entropy $\mathcal{H}(s' \mid s, z)$. Intuitively, skills $z$ can be interpreted as abstracted action sequences which are identifiable by the transitions generated in the environment (and not just by the current state). Thus, optimizing this mutual information can be understood as encoding a diverse set of skills in the latent space $\mathcal{Z}$, while making the transitions for a given $z \in \mathcal{Z}$ predictable. We use the entropy-decomposition in Equation 2 to connect this objective with model-based control.

We want to optimize the our skill-conditioned controller $\pi(a \mid s, z)$ such that the latent space $z \sim p(z)$ is maximally informative about the transitions $s \to s'$. Using the definition of conditional mutual information, we can rewrite Equation 2 as:

$$\mathcal{I}(s'; z \mid s) = \int p(z, s, s') \log \frac{p(s' \mid s, z)}{p(s' \mid s)} ds' ds dz \tag{3}$$

We assume the following generative model: $p(z, s, s') = p(z)p(s \mid z)p(s' \mid s, z)$, where $p(z)$ is user specified prior over $\mathcal{Z}$, $p(s|z)$ denotes the stationary state-distribution induced by $\pi(a \mid s, z)$ for a skill $z$ and $p(s' \mid s, z)$ denotes the transition distribution under skill $z$. Note, $p(s' \mid s, z) = \int p(s' \mid s, a)\pi(a \mid s, z)da$ is intractable to compute because the underlying dynamics are unknown. However, we can variationally lower bound the objective as follows:

$$\mathcal{I}(s'; z \mid s) = \mathbb{E}_{z,s,s' \sim p}\left[ \log \frac{p(s' \mid s, z)}{p(s' \mid s)} \right]$$

$$= \mathbb{E}_{z,s,s' \sim p}\left[ \log \frac{q_\phi(s' \mid s, z)}{p(s' \mid s)} \right] + \mathbb{E}_{s,z \sim p}\left[ \mathcal{D}_{KL}(p(s' \mid s, z) \mid\mid q_\phi(s' \mid s, z)) \right]$$

$$\geq \mathbb{E}_{z,s,s' \sim p}\left[ \log \frac{q_\phi(s' \mid s, z)}{p(s' \mid s)} \right] \tag{4}$$

where we have used the non-negativity of KL-divergence, that is $\mathcal{D}_{KL} \geq 0$. Note, skill-dynamics $q_\phi$ represents the variational approximation for the transition function $p(s' \mid s, z)$, which enables model-based control as described in Section 4. Equation 4 suggests an alternating optimization between $q_\phi$ and $\pi$, summarized in Algorithm 1. In every iteration:

(*Tighten variational lower bound*) We minimize $D_{KL}(p(s' \mid s, z) \mid\mid q_\phi(s' \mid s, z))$ with respect to the parameters $\phi$ on $z, s \sim p$ to tighten the lower bound. For general function approximators like neural networks, we can write the gradient for $\phi$ as follows:

$$\nabla_\phi \mathbb{E}_{s,z}[\mathcal{D}_{KL}(p(s' \mid s, z) \mid\mid q_\phi(s' \mid s, z))] = \nabla_\phi \mathbb{E}_{z,s,s'}\left[\log \frac{p(s' \mid s, z)}{q_\phi(s' \mid s, z)}\right]$$

$$= -\mathbb{E}_{z,s,s'}\left[\nabla_\phi \log q_\phi(s' \mid s, z)\right] \quad (5)$$

which corresponds to maximizing the likelihood of the samples from $p$ under $q_\phi$.

(*Maximize approximate lower bound*) After fitting $q_\phi$, we can optimize $\pi$ to maximize $\mathbb{E}_{z,s,s'}[\log q_\phi(s' \mid s, z) - \log p(s' \mid s)]$. Note, this is a reinforcement-learning style optimization with a reward function $\log q_\phi(s' \mid s, z) - \log p(s' \mid s)$. However, $\log p(s' \mid s)$ is intractable to compute, so we approximate the reward function for $\pi$:

$$r_z(s, a, s') = \log \frac{q_\phi(s' \mid s, z)}{\sum_{i=1}^{L} q_\phi(s' \mid s, z_i)} + \log L, \quad z_i \sim p(z). \quad (6)$$

The approximation is motivated as follows: $p(s' \mid s) = \int p(s' \mid s, z)p(z|s)dz \approx \int q_\phi(s' \mid s, z)p(z)dz \approx \frac{1}{L}\sum_{i=1}^{L} q_\phi(s' \mid s, z_i)$ for $z_i \sim p(z)$, where $L$ denotes the number of samples from the prior. We are using the marginal of variational approximation $q_\phi$ over the prior $p(z)$ to approximate the marginal distribution of transitions. We discuss this approximation in Appendix C. Note, the final reward function $r_z$ encourages the policy $\pi$ to produce transitions that are (a) predictable under $q_\phi$ (*predictability*) and (b) different from the transitions produced under $z_i \sim p(z)$ (*diversity*).

To generate samples from $p(z, s, s')$, we use the rollouts from the current policy $\pi$ for multiple samples $z \sim p(z)$ in an episodic setting for a fixed horizon $T$. We also introduce entropy regularization for $\pi(a \mid s, z)$, which encourages the policy to discover action-sequences with similar state-transitions and to be clustered under the same skill $z$, making the policy robust besides encouraging exploration (Haarnoja et al., 2018a). The use of entropy regularization can be justified from an information bottleneck perspective as discussed for Information Maximization algorithm in (Mohamed & Rezende, 2015). This is even more extensively discussed from the graphical model perspective in Appendix B, which connects unsupervised skill discovery and information bottleneck literature, while also revealing the temporal nature of skills $z$. Details corresponding to implementation and hyperparameters are discussed in Appendix A.

## 4 Planning using Skill Dynamics

Given the learned skills $\pi(a \mid s, z)$ and their respective skill-transition dynamics $q_\phi(s' \mid s, z)$, we can perform model-based planning in the latent space $\mathcal{Z}$ to optimize for a reward $r$ that is given to the agent at test time. Note, that this essentially allows us to perform zero-shot planning given the unsupervised pre-training procedure described in Section 3.

In order to perform planning, we employ the model-predictive-control (MPC) paradigm Garcia et al. (1989), which in a standard setting generates a set of action plans $P_k = (a_{k,1}, \ldots a_{k,H}) \sim P$ for a planning horizon $H$. The MPC plans can be generated due to the fact that the planner is able to simulate the trajectory $\hat{\tau}_k = (s_{k,1}, a_{k,1} \ldots s_{k,H+1})$ assuming access to the transition dynamics $\hat{p}(s' \mid s, a)$. In addition, each plan computes the reward $r(\hat{\tau}_k)$ for its trajectory according to the reward function $r$ that is provided for the test-time task. Following the MPC principle, the planner selects the best plan according to the reward function $r$ and executes its first action $a_1$. The planning algorithm repeats this procedure for the next state iteratively until it achieves its goal.

We use a similar strategy to design an MPC planner to exploit previously-learned skill-transition dynamics $q_\phi(s' \mid s, z)$. Note that unlike conventional model-based RL, we generate a plan $P_k = (z_{k,1}, \ldots z_{k,H_P})$ in the latent space $\mathcal{Z}$ as opposed to the action space $\mathcal{A}$ that would be used by a standard planner. Since the primitives are temporally meaningful, it is beneficial to hold a primitive

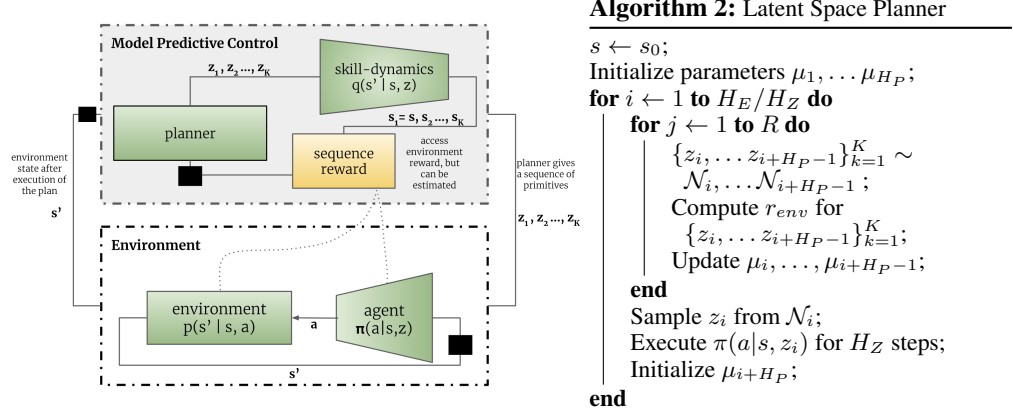

**Algorithm 2:** Latent Space Planner

$s \leftarrow s_0$;
Initialize parameters $\mu_1, \ldots \mu_{H_P}$;
**for** $i \leftarrow 1$ **to** $H_E/H_Z$ **do**
  **for** $j \leftarrow 1$ **to** $R$ **do**
    $\{z_i, \ldots z_{i+H_P-1}\}_{k=1}^K \sim$ $\mathcal{N}_i, \ldots \mathcal{N}_{i+H_P-1}$ ;
    Compute $r_{env}$ for $\{z_i, \ldots z_{i+H_P-1}\}_{k=1}^K$;
    Update $\mu_i, \ldots, \mu_{i+H_P-1}$;
  **end**
  Sample $z_i$ from $\mathcal{N}_i$;
  Execute $\pi(a|s, z_i)$ for $H_Z$ steps;
  Initialize $\mu_{i+H_P}$;
**end**

Figure 3: At test time, the planner executes simulates the transitions in environment using skill-dynamics $q$, and updates the distribution of plans according to the computed reward on the simulated trajectories. After a few updates to the plan, the first primitive is executed in the environment using the learned agent $\pi$.

for a horizon $H_Z > 1$, unlike actions which are usually held for a single step. Thus, effectively, the planning horizon for our latent space planner is $H = H_P \times H_Z$, enabling longer-horizon planning using fewer primitives. Similar to the standard MPC setting, the latent space planner simulates the trajectory $\hat{\tau}_k = (s_{k,1}, z_{k,1}, a_{k,1}, s_{k,2}, z_{k,2}, a_{k,2}, \ldots s_{k,H+1})$ and computes the reward $r(\hat{\tau}_k)$. After a small number of trajectory samples, the planner selects the first latent action $z_1$ of the best plan, executes it for $H_Z$ steps in the environment, and the repeats the process until goal completion.

The latent planner $P$ maintains a distribution of latent plans, each of length $H_P$. Each element in the sequence represents the distribution of the primitive to be executed at that time step. For continuous spaces, each element of the sequence can be modelled using a normal distribution, $\mathcal{N}(\mu_1, \Sigma), \ldots \mathcal{N}(\mu_{H_P}, \Sigma)$. We refine the planning distributions for $R$ steps, using $K$ samples of latent plans $P_k$, and compute the $r_k$ for the simulated trajectory $\hat{\tau}_k$. The update for the parameters follows that in Model Predictive Path Integral (MPPI) controller Williams et al. (2016):

$$\mu_i = \sum_{k=1}^K \frac{\exp(\gamma r_k)}{\sum_{p=1}^K \exp(\gamma r_p)} z_{k,i} \quad \forall i = 1, \ldots H_P \tag{7}$$

While we keep the covariance matrix of the distributions fixed, it is possible to update that as well as shown in Williams et al. (2016). We show an overview of the planning algorithm in Figure 3, and provide more implementation details in Appendix A.

## 5 RELATED WORK

Central to our method is the concept of skill discovery via mutual information maximization. This principle, proposed in prior work that utilized purely model-free unsupervised RL methods (Daniel et al., 2012; Florensa et al., 2017; Eysenbach et al., 2018; Gregor et al., 2016; Warde-Farley et al., 2018; Thomas et al., 2018), aims to learn diverse skills via a discriminability objective: a good set of skills is one where it is easy to distinguish the skills from each other, which means they perform distinct tasks and cover the space of possible behaviors. Building on this prior work, we distinguish our skills based on how they modify the original uncontrolled dynamics of the system. This simultaneously encourages the skills to be both *diverse* and *predictable*. We also demonstrate that constraining the skills to be predictable makes them more amenable for hierarchical composition and thus, more useful on downstream tasks.

Another line of work that is conceptually close to our method copes with intrinsic motivation (Oudeyer & Kaplan, 2009; Oudeyer et al., 2007; Schmidhuber, 2010) which is used to drive the agent's exploration. Examples of such works include empowerment Klyubin et al. (2005); Mohamed & Rezende (2015), count-based exploration Bellemare et al. (2016); Oh et al. (2015); Tang et al. (2017); Fu et al. (2017), information gain about agent's dynamics Stadie et al. (2015) and

forward-inverse dynamics models Pathak et al. (2017). While our method uses an information-theoretic objective that is similar to these approaches, it is used to learn a variety of skills that can be directly used for model-based planning, which is in contrast to learning a better exploration policy for a single skill.

The skills discovered using our approach can also provide extended actions and temporal abstraction, which enable more efficient exploration for the agent to solve various tasks, reminiscent of hierarchical RL (HRL) approaches. This ranges from the classic option-critic architecture (Sutton et al., 1999; Stolle & Precup, 2002; Perkins et al., 1999) to some of the more recent work (Bacon et al., 2017; Vezhnevets et al., 2017; Nachum et al., 2018; Hausman et al., 2018). However, in contrast to end-to-end HRL approaches (Heess et al., 2016; Peng et al., 2017), we can leverage a stable, two-phase learning setup. The primitives learned through our method provide action and temporal abstraction, while planning with skill-dynamics enables hierarchical composition of these primitives, bypassing many problems of end-to-end HRL.

In the second phase of our approach, we use the learned skill-transition dynamics models to perform model-based planning - an idea that has been explored numerous times in the literature. Model-based reinforcement learning has been traditionally approached with methods that are well-suited for low-data regimes such as Gaussian Processes (Rasmussen, 2003) showing significant data-efficiency gains over model-free approaches (Deisenroth et al., 2013; Kamthe & Deisenroth, 2017; Kocijan et al., 2004; Ko et al., 2007). More recently, due to the challenges of applying these methods to high-dimensional state spaces, MBRL approaches employs Bayesian deep neural networks (Nagabandi et al., 2018; Chua et al., 2018b; Gal et al., 2016; Fu et al., 2016; Lenz et al., 2015) to learn dynamics models. In our approach, we take advantage of the deep dynamics models that are conditioned on the skill being executed, simplifying the modelling problem. In addition, the skills themselves are being learned with the objective of being predictable, further assists with the learning of the dynamics model. There also have been multiple approaches addressing the planning component of MBRL including linear controllers for local models (Levine et al., 2016; Kumar et al., 2016; Chebotar et al., 2017), uncertainty-aware (Chua et al., 2018b; Gal et al., 2016) or deterministic planners (Nagabandi et al., 2018) and stochastic optimization methods (Williams et al., 2016). The main contribution of our work lies in discovering model-based skill primitives that can be further combined by a standard model-based planner, therefore we take advantage of an existing planning approach - Model Predictive Path Integral (Williams et al., 2016) that can leverage our pre-trained setting.

# 6 EXPERIMENTS

Through our experiments, we aim to demonstrate that: (a) DADS as a general purpose skill discovery algorithm can scale to high-dimensional problems; (b) discovered skills are amenable to hierarchical composition and; (c) not only is planning in the learned latent space feasible, but it is competitive to strong baselines. In Section 6.1, we provide visualizations and qualitative analysis of the skills learned using DADS. We demonstrate in Section 6.2 and Section 6.4 that optimizing the primitives for predictability renders skills more amenable to temporal composition that can be used for Hierarchical RL. We benchmark against state-of-the-art model-based RL baseline in Section 6.3, and against goal-conditioned RL in Section 6.5.

## 6.1 QUALITATIVE ANALYSIS

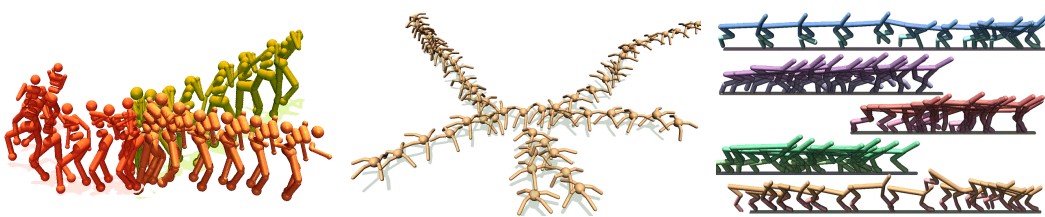

Figure 4: Skills learned on different MuJoCo environments in the OpenAI gym. DADS can discover diverse skills without any extrinsic rewards, even for problems with high-dimensional state and action spaces.

In this section, we provide a qualitative discussion of the unsupervised skills learned using DADS. We use the MuJoCo environments (Todorov et al., 2012) from the OpenAI gym as our test-bed (Brockman et al., 2016). We find that our proposed algorithm can learn diverse skills without any reward, even in problems with high-dimensional state and actuation, as illustrated in Figure 4. DADS can discover primitives for Half-Cheetah to run forward and backward with multiple different gaits, for Ant to navigate the environment using diverse locomotion primitives and for Humanoid to walk using stable locomotion primitives with diverse gaits and direction. The videos of the discovered primitives are available at: `https://sites.google.com/view/dads-skill`

Qualitatively, we find the skills discovered by DADS to be predictable and stable, in line with implicit constraints of the proposed objective. While the Half-Cheetah will learn to run in both backward and forward directions, DADS will disincentivize skills which make Half-Cheetah flip owing to the reduced predictability on landing. Similarly, skills discovered for Ant rarely flip over, and tend to provide stable navigation primitives in the environment. This also incentivizes the Humanoid, which is characteristically prone to collapsing and extremely unstable by design, to discover gaits which are stable for sustainable locomotion.

One of the significant advantages of the proposed objective is that it is compatible with continuous skill spaces, which has not been shown in prior work on skill discovery (Eysenbach et al., 2018). Not only does this allow us to embed a large and diverse set of skills into a compact latent space, but also the smoothness of the learned space allows us to interpolate between behaviors generated in the environment. We demonstrate this on the Ant environment (Figure 5), where we learn two-dimensional continuous skill space with a uniform prior over $(-1, 1)$ in each dimension, and compare it to a discrete skill space with a uniform prior over 20 skills. Similar to Eysenbach et al. (2018), we restrict the observation space of the skill-dynamics $q$ to the cartesian coordinates $(x, y)$. We hereby call this the *x-y prior*, and discuss its role in Section 6.2.

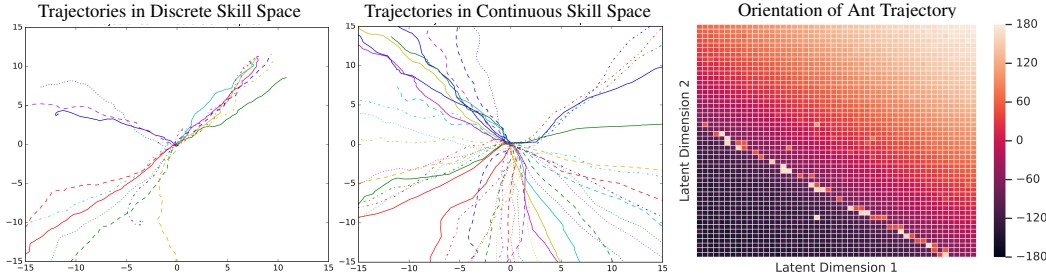

Figure 5: (Left, Centre) X-Y traces of Ant skills and (Right) Heatmap to visualize the learned continuous skill space. Traces demonstrate that the continuous space offers far greater diversity of skills, while the heatmap demonstrates that the learned space is smooth, as the orientation of the X-Y trace varies smoothly as a function of the skill.

In Figure 5, we project the trajectories of the learned Ant skills from both discrete and continuous spaces onto the Cartesian plane. From the traces of the skills, it is clear that the continuous latent space can generate more diverse trajectories. We demonstrate in Section 6.3, that continuous primitives are more amenable to hierarchical composition and generally perform better on downstream tasks. More importantly, we observe that the learned skill space is semantically meaningful. The heatmap in Figure 5 shows the orientation of the trajectory (with respect to the $x$-axis) as a function of the skill $z \in \mathcal{Z}$, which varies smoothly as $z$ is varied, with explicit interpolations shown in Appendix D.

## 6.2 SKILL VARIANCE ANALYSIS

In an unsupervised skill learning setup, it is important to optimize the primitives to be diverse. However, we argue that diversity is not sufficient for the learned primitives to be useful for downstream tasks. Primitives must exhibit low-variance behavior, which enables long-horizon composition of the learned skills in a hierarchical setup. We analyze the variance of the $x$-$y$ trajectories in the environment, where we also benchmark the variance of the primitives learned by DIAYN (Eysenbach et al., 2018). For DIAYN, we use the $x$-$y$ prior for the skill-discriminator, which biases the discovered skills to diversify in the $x$-$y$ space. This step was necessary for that baseline to obtain a

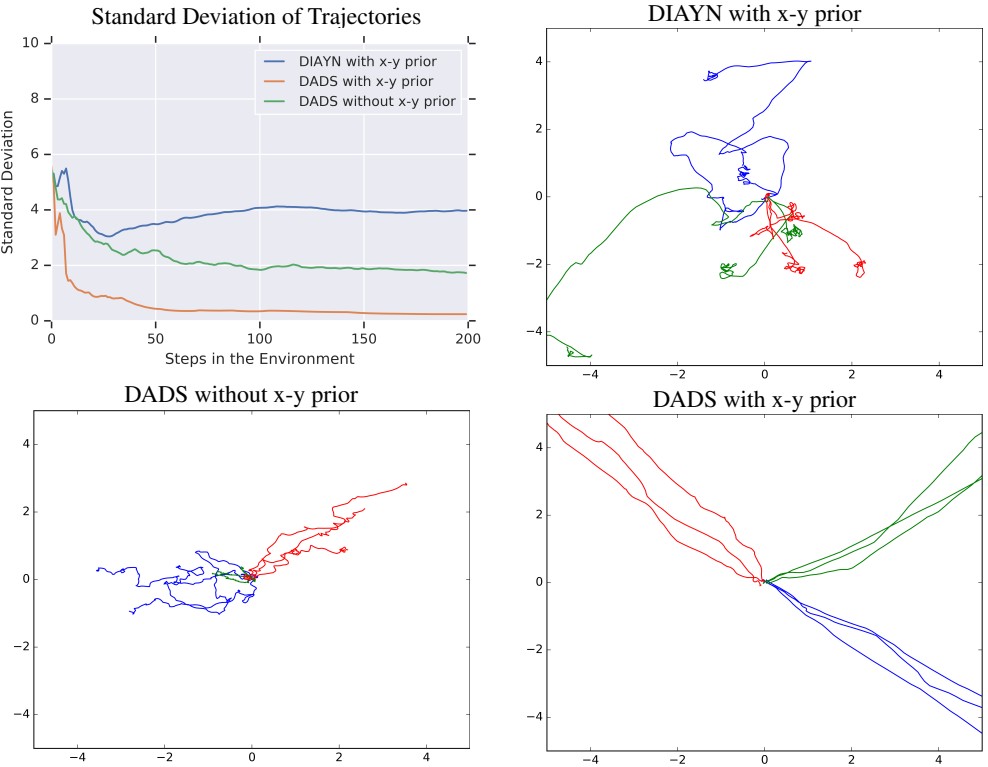

Figure 6: (Top-Left) Standard deviation of Ant's position as a function of steps in the environment, averaged over multiple skills and normalized by the norm of the position. (Top-Right to Bottom-Left Clockwise) X-Y traces of skills learned using DIAYN with $x$-$y$ prior, DADS with $x$-$y$ prior and DADS without x-y prior, where the same color represents trajectories resulting from the execution of the same skill $z$ in the environment. High variance skills from DIAYN offer limited utility for hierarchical control.

competitive set of navigation skills. Figure 6 (Top-Left) demonstrates that DADS, which optimizes the primitives for predictability and diversity, yields significantly lower-variance primitives when compared to DIAYN, which only optimizes for diversity. This is starkly demonstrated in the plots of X-Y traces of skills learned in different setups. Skills learned by DADS show significant control over the trajectories generated in the environment, while skills from DIAYN exhibit high variance in the environment, which limits their utility for hierarchical control. This is further demonstrated quantitatively in Section 6.4.

While optimizing for predictability already significantly reduces the variance of the trajectories generated by a primitive, we find that using the $x$-$y$ prior with DADS brings down the skill variance even further. For quantitative benchmarks in the next sections, we assume that the Ant skills are learned using an $x$-$y$ prior on the observation space, for both DADS and DIAYN.

## 6.3 MODEL-BASED REINFORCEMENT LEARNING

The key utility of learning a parametric model $q_\phi(s'|s, z)$ is to take advantage of planning algorithms for downstream tasks, which can be extremely sample-efficient. In our setup, we can solve test-time tasks in zero-shot, that is *without any learning on the downstream task*. We compare with the state-of-the-art model-based RL method (Chua et al., 2018a), which learns a dynamics model parameterized as $p(s'|s, a)$, on the task of the Ant navigating to a specified goal with a dense reward. Given a goal $g$, reward at any position $u$ is given by $r(u) = -\|g - u\|_2$. We benchmark our method against the following variants:

- Random-MBRL (*rMBRL*): We train the model $p(s'|s, a)$ on randomly collected trajectories, and test the zero-shot generalization of the model on a distribution of goals.

- Weak-oracle MBRL (*WO-MBRL*): We train the model $p(s'|s, a)$ on trajectories generated by the planner to navigate to a goal, randomly sampled in every episode. The distribution of goals during training matches the distribution at test time.
- Strong-oracle MBRL (*SO-MBRL*): We train the model $p(s'|s, a)$ on a trajectories generated by the planner to navigate to a specific goal, which is fixed for both training and test time.

Amongst the variants, only the rMBRL matches our assumptions of having an unsupervised task-agnostic training. Both WO-MBRL and SO-MBRL benefit from goal-directed exploration during training, a significant advantage over DADS, which only uses mutual-information-based exploration.

We use $\Delta = \sum_{t=1}^{H} \frac{-r(u)}{H\|g\|_2}$ as the metric, which represents the distance to the goal $g$ averaged over the episode (with the same fixed horizon $H$ for all models and experiments), normalized by the initial distance to the goal $g$. Therefore, lower $\Delta$ indicates better performance and $0 < \Delta \leq 1$ (assuming the agent goes closer to the goal). The test set of goals is fixed for all the methods, sampled from $[-15, 15]^2$.

Figure 7 demonstrates that the zero-shot planning significantly outperforms all model-based RL baselines, despite the advantage of the baselines being trained on the test goal(s). For the experiment depicted in Figure 7 (Right), DADS has an unsupervised pre-training phase, unlike SO-MBRL which is training directly for the task. A comparison with Random-MBRL shows the significance of mutual-information-based exploration, especially with the right parameterization and priors. This experiment also demonstrates the advantage of learning a continuous space of primitives, which outperforms planning on discrete primitives.

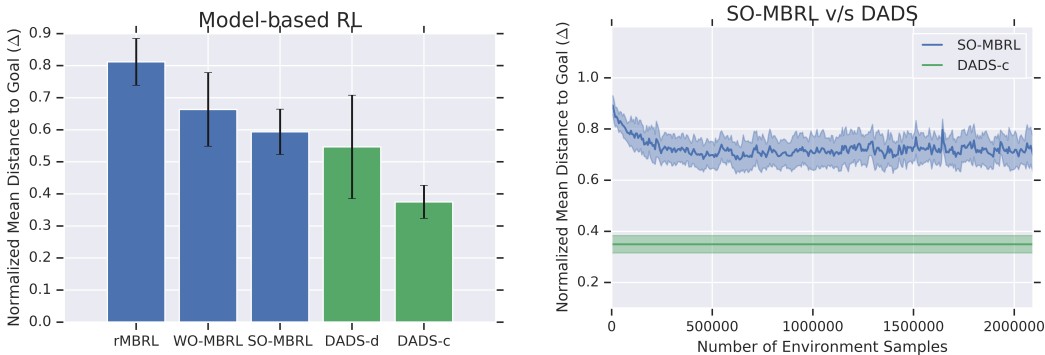

Figure 7: (Left) The results of the MPPI controller on skills learned using DADS-c (continuous primitives) and DADS-d (discrete primitives) significantly outperforms state-of-the-art model-based RL. (Right) Planning for a new task does not require any additional training and outperforms model-based RL being trained for the specific task.

## 6.4 HIERARCHICAL CONTROL WITH UNSUPERVISED PRIMITIVES

We benchmark hierarchical control for primitives learned without supervision, against our proposed scheme using an MPPI based planner on top of DADS-learned skills. We persist with the task of Ant-navigation as described in 6.3. We benchmark against Hierarchical DIAYN (Eysenbach et al., 2018), which learns the skills using the DIAYN objective, freezes the low-level policy and learns a meta-controller that outputs the skill to be executed for the next $H_Z$ steps. We provide the $x$-$y$ prior to the DIAYN's discriminator while learning the skills for the Ant agent. The performance of the meta-controller is constrained by the low-level policy, however, this hierarchical scheme is agnostic to the algorithm used to learn the low-level policy. To contrast the quality of primitives learned by the DADS and DIAYN, we also benchmark against Hierarchical DADS, which learns a meta-controller the same way as Hierarchical DIAYN, but learns the skills using DADS.

From Figure 8 (Left) We find that the meta-controller is unable to compose the skills learned by DIAYN, while the same meta-controller can learn to compose skills by DADS to navigate the Ant to different goals. This result seems to confirm our intuition described in Section 6.2, that the high variance of the DIAYN skills limits their temporal compositionality. Interestingly, learning a RL

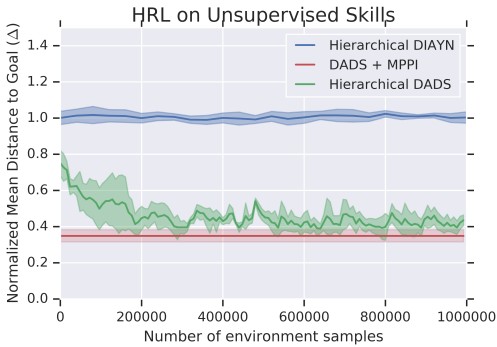 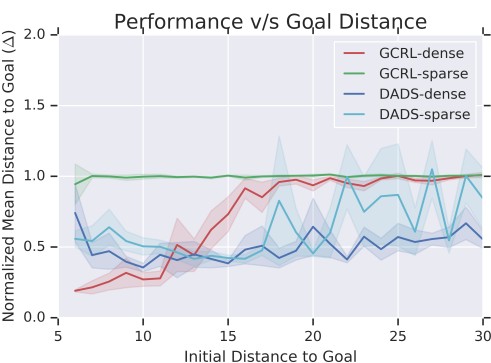

Figure 8: (Left) A RL-trained meta-controller is unable to compose primitive learned by DIAYN to navigate Ant to a goal, while it succeeds to do so using the primitives learned by DADS. (Right) Goal-Conditioned RL (GCRL-dense/sparse) does not generalize outside its training distribution, while MPPI controller on learned skills (DADS-dense/sparse) experiences significantly smaller degrade in performance.

meta-controller reaches similar performance to the MPPI controller, taking an additional $200,000$ samples per goal.

### 6.5 GOAL-CONDITIONED RL

To demonstrate the benefits of our approach over model-free RL, we benchmark against goal-conditioned RL on two versions of Ant-navigation: (a) with a dense reward $r(u)$ and (b) with a sparse reward $r(u) = 1$ if $\|u - g\|_2 \leq \epsilon$, else 0. We train the goal-conditioned RL agent using soft actor-critic, where the state variable of the agent is augmented with $u - g$, the position delta to the goal. The agent gets a randomly sampled goal from $[-10, 10]^2$ at the beginning of the episode.

In Figure 8 (Right), we measure the average performance of the all the methods as a function of the initial distance of the goal, ranging from 5 to 30 metres. For dense reward navigation, we observe that while model-based planning on DADS-learned skills degrades smoothly as the initial distance to goal to increases, goal-conditioned RL experiences a sudden deterioration outside the goal distribution it was trained on. Even within the goal distribution observed during training of goal-conditioned RL model, skill-space planning performs competitively to it. With sparse reward navigation, goal-conditioned RL is unable to navigate, while MPPI demonstrates comparable performance to the dense reward up to about 20 metres. This highlights the utility of learning task-agnostic skills, which makes them more general while showing that latent space planning can be leveraged for tasks requiring long-horizon planning.

## 7 CONCLUSION

We have proposed a novel unsupervised skill learning algorithm that is amenable to model-based planning for hierarchical control on downstream tasks. We show that our skill learning method can scale to high-dimensional state-spaces, while discovering a diverse set of low-variance skills. In addition, we demonstrated that, without any training on the specified task, we can compose the learned skills to outperform competitive model-based baselines that were trained with the knowledge of the test tasks. We plan to extend the algorithm to work with off-policy data, potentially using relabelling tricks (Andrychowicz et al., 2017; Nachum et al., 2018) and explore more nuanced planning algorithms. We plan to apply the hereby-introduced method to different domains, such as manipulation and enable skill/model discovery directly from images.

## 8 ACKNOWLEDGEMENTS

We would like to thank Evan Liu, Ben Eysenbach, Anusha Nagabandi for their help in reproducing the baselines for this work. We are thankful to Ben Eysenbach for their comments and discussion on the initial drafts. We would also like to acknowledge Ofir Nachum, Alex Alemi, Daniel Freeman, Yiding Jiang, Allan Zhou and other colleagues at Google Brain for their helpful feedback and discussions at various stages of this work. We are also thankful to Michael Ahn and others in Adept team for their support, especially with the infrastructure setup and scaling up the experiments.

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

# A  IMPLEMENTATION DETAILS

All of our models are written in the open source Tensorflow-Agents (Sergio Guadarrama, Anoop Korattikara, Oscar Ramirez, Pablo Castro, Ethan Holly, Sam Fishman, Ke Wang, Ekaterina Gonina, Chris Harris, Vincent Vanhoucke, Eugene Brevdo, 2018), based on Tensorflow (Abadi et al., 2015).

## A.1  SKILL SPACES

When using discrete spaces, we parameterize $\mathcal{Z}$ as one-hot vectors. These one-hot vectors are randomly sampled from the uniform prior $p(z) = \frac{1}{D}$, where $D$ is the number of skills. We experiment with $D \leq 128$. For discrete skills learnt for MuJoCo Ant in Section 6.3, we use $D = 20$. For continuous spaces, we sample $z \sim \text{Uniform}(-1, 1)^D$. We experiment with $D = 2$ for Ant learnt with x-y prior, $D = 3$ for Ant learnt without x-y prior (that is full observation space), to $D = 5$ for Humanoid on full observation spaces. The skills are sampled once in the beginning of the episode and fixed for the rest of the episode. However, it is possible to resample the skill from the prior within the episode, which allows for every skill to experience a different distribution than the initialization distribution. This also encourages discovery of skills which can be composed temporally. However, this increases the hardness of problem, especially if the skills are re-sampled from the prior frequently.

## A.2  AGENT

We use SAC as the optimizer for our agent $\pi(a \mid s, z)$, in particular, EC-SAC (Haarnoja et al., 2018b). The $s$ input to the policy generally excludes global co-ordinates $(x, y)$ of the centre-of-mass, available for a lot of enviroments in OpenAI gym, which helps produce skills agnostic to the location of the agent. We restrict to two hidden layers for our policy and critic networks. However, to improve the expressivity of skills, it is beneficial to increase the capacity of the networks. The hidden layer sizes can vary from $(128, 128)$ for Half-Cheetah to $(512, 512)$ for Ant and $(1024, 1024)$ for Humanoid. The critic $Q(s, a, z)$ is similarly parameterized. The target function for critic $Q$ is updated every iteration using a soft updates with co-efficient of $0.005$. We use Adam (Kingma & Ba, 2014) optimizer with a fixed learning rate of $3e - 4$ , and a fixed initial entropy co-efficient $\beta = 0.1$. While the policy is parameterized as a normal distribution $\mathcal{N}(\mu(s, z), \Sigma(s, z))$ where $\Sigma$ is a diagonal covariance matrix, it undergoes through tanh transformation, to transform the output to the range $(-1, 1)$ and constrain to the action bounds.

## A.3 SKILL-DYNAMICS

Skill-dynamics, denoted by $q(s' \mid s, z)$, is parameterized by a deep neural network. A common trick in model-based RL is to predict the $\Delta s = s' - s$, rather than the full state $s'$. Hence, the prediction network is $q(\Delta s \mid s, z)$. Note, both parameterizations can represent the same set of functions. However, the latter will be easy to learn as $\Delta s$ will be centred around 0. We exclude the global co-ordinates from from the state input to $q$. However, we can (and we still do) predict $\Delta_x, \Delta_y$, because reward functions for goal-based navigation generally rely on the position prediction from the model. This represents another benefit of predicting state-deltas, as we can still predict changes in position without explicitly knowing the global position.

The output distribution is modelled as a Mixture-of-Experts (Jacobs et al., 1991). We fix the number of experts to be 4. We model each expert as a Gaussian distribution. The input $(s, z)$ goes through two hidden layers (the same capacity as that of policy and critic networks, for example $(512, 512)$ for Ant). The output of the two hidden layers is used as an input to the mixture-of-experts, which is linearly transformed to output the parameters of the Gaussian distribution, and a discrete distribution over the experts using a softmax distribution. In practice, we fix the covariance matrix of the Gaussian experts to be an identity matrix, so we only need to output the means for the experts. We use batch-normalization for both input and the hidden layers. We normalize the output targets using their batch-average and batch-standard deviation, similar to batch-normalization.

## A.4 OTHER HYPERPARAMETERS

The episode horizon is generally kept shorter for stable agents like Ant (200), while longer for unstable agents like Humanoid (1000). For Ant, longer episodes do not add value, but Humanoid can benefit from longer episodes as it helps it filter skills which are unstable. The optimization scheme is on-policy, and we collect 2000 steps for Ant and 4000 steps for Humanoid in one iteration. The intuition is to experience trajectories generated by multiple skills (approximately 10) in a batch. Re-sampling skills can enable experiencing larger number of skills. Once a batch of episodes is collected, the skill-dynamics is updated using Adam optimizer with a fixed learning rate of $3e - 4$. The batch size is 128, and we carry out 32 steps of gradient descent. To compute the intrinsic reward, we need to resample the prior for computing the denominator. For continuous spaces, we set $L = 500$. For discrete spaces, we can marginalize over all skills. After the intrinsic reward is computed, the policy and critic networks are updated for 128 steps with a batch size of 128. The intuition is to ensure that every sample in the batch is seen for policy and critic updates about $3 - 4$ times in expectation.

## A.5 PLANNING AND EVALUATION SETUPS

For evaluation, we fix the episode horizon to 200 for all models in all evaluation setups. Depending upon the size of the latent space and planning horizon, the number of samples from the planning distribution $P$ is varied between $10 - 200$. For $H_P = 1, H_Z = 10$ and a $2D$ latent space, we use 50 samples from the planning distribution $P$. The co-efficient $\gamma$ for MPPI is fixed to 10. We use a setting of $H_P = 1$ and $H_Z = 10$ for dense-reward navigation, in which case we set the number of refine steps $R = 10$. However, for sparse reward navigation it is important to have a longer horizon planning, in which case we set $H_P = 4, H_Z = 25$ with a higher number of samples from the planning distribution (200 from $P$). Also, when using longer planning horizons, we found that smoothing the sampled plans help. Thus, if the sampled plan is $z_1, z_2, z_3, z_4 \ldots$, we smooth the plan to make $z_2 = \beta z_1 + (1 - \beta) z_2$ and so on, with $\beta = 0.9$.

For hierarchical controllers being learnt on top of low-level unsupervised primitives, we use PPO (Schulman et al., 2017) for discrete action skills, while we use SAC for continuous skills. We keep the number of steps after which the meta-action is decided as 10 (that is $H_Z = 10$). The hidden layer sizes of the meta-controller are $(128, 128)$. We use a learning rate of $1e - 4$ for PPO and $3e - 4$ for SAC.

For our model-based RL baseline PETS, we use an ensemble size of 3, with a fixed planning horizon of 20. For the model, we use a neural network with two hidden layers of size 400. In our experiments, we found that MPPI outperforms CEM, so we report the results using the MPPI as our controller.

## B  GRAPHICAL MODELS, INFORMATION BOTTLENECK AND UNSUPERVISED SKILL LEARNING

We now present a novel perspective on unsupervised skill learning, motivated from the literature on information bottleneck. This section takes inspiration from (Alemi & Fischer, 2018), which helps us provide a rigorous justification for our objective proposed earlier. To obtain our unsupervised RL objective, we setup a graphical model $P$ as shown in Figure 9, which represents the distribution of trajectories generated by a given policy $\pi$. The joint distribution is given by:

$$p(s_1, a_1 \ldots a_{T-1}, s_T, z) = p(z)p(s_1) \prod_{t=1}^{T-1} \pi(a_t|s_t, z)p(s_{t+1}|s_t, a_t). \tag{8}$$

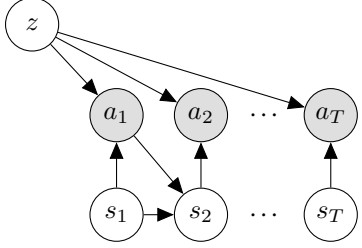

Figure 9: Graphical model for the world $P$ in which the trajectories are generated while interacting with the environment. Shaded nodes represent the distributions we optimize.

Figure 10: Graphical model for the world $N$ which is the desired representation of the world.

We setup another graphical model $N$, which represents the desired model of the world. In particular, we are interested in approximating $p(s'|s, z)$, which represents the transition function for a particular primitive. This abstraction helps us get away from knowing the exact actions, enabling model-based planning in behavior space (as discussed in the main paper). The joint distribution for $N$ shown in Figure 10 is given by:

$$\eta(s_1, a_1, \ldots s_T, a_T, z) = \eta(z)\eta(s_1) \prod_{t=1}^{T-1} \eta(a_t)\eta(s_{t+1}|s_t, z). \tag{9}$$

The goal of our approach is to optimize the distribution $\pi(a|s, z)$ in the graphical model $P$ to minimize the distance between the two distributions, when transforming to the representation of the graphical model $Z$. In particular, we are interested in minimizing the KL divergence between $p$ and $\eta$, that is $\mathcal{D}_{KL}(p||\eta)$. Note, if $N$ had the same structure as $P$, the information lost in projection would be $0$ for any valid $P$. Interestingly, we can exploit the following result from in Friedman et al. (2001) to setup the objective for $\pi$, without explicitly knowing $\eta$:

$$\min_{\eta} \mathcal{D}_{KL}(p||\eta) = \mathcal{I}_P - \mathcal{I}_N, \tag{10}$$

where $\mathcal{I}_P$ and $\mathcal{I}_N$ represents the multi-information for distribution $P$ on the respective graphical models. Note, $\min_{\eta \in N} \mathcal{D}_{KL}(p||\eta)$, which is the reverse information projection (Csiszár & Matus, 2003). The multi-information (Slonim et al., 2005) for a graphical model $G$ with nodes $g_i$ is defined as:

$$\mathcal{I}_G = \sum_i I(g_i; Pa(g_i)), \tag{11}$$

where $Pa(g_i)$ denotes the nodes upon which $g_i$ has direct conditional dependence in $G$. Using this definition, we can compute the multi-information terms:

$$\mathcal{I}_P = \sum_{t=1}^{T} I(a_t; \{s_t, z\}) + \sum_{t=2}^{T} I(s_t; \{s_{t-1}, a_{t-1}\}) \quad \text{and} \quad \mathcal{I}_N = \sum_{t=2}^{T} I(s_t; \{s_{t-1}, z\}). \tag{12}$$

Following the Optimal Frontier argument in (Alemi & Fischer, 2018), we introduce Lagrange multipliers $\beta_t \geq 0, \delta_t \geq 0$ for the information terms in $\mathcal{I}_P$ to setup an objective $R(\pi)$ to be maximized with respect to $\pi$:

$$R(\pi) = \sum_{t=1}^{T-1} I(s_{t+1}; \{s_t, z\}) - \beta_t I(a_t; \{s_t, z\}) - \delta_t \mathcal{I}(s_{t+1}; \{s_t, a_t\}) \tag{13}$$

$$\tag{14}$$

As the underlying dynamics are fixed and unknown, we simplify the optimization by setting $\delta_t = 0$ which intuitively corresponds to us neglecting the unchangeable information of the underlying dynamics. This gives us

$$R(\pi) = \sum_{t=1}^{T-1} I(s_{t+1}; \{s_t, z\}) - \beta_t I(a_t; \{s_t, z\}) \tag{15}$$

$$\geq \sum_{t=1}^{T-1} I(s_{t+1}; z \mid s_t) - \beta_t I(a_t; \{s_t, z\}) \tag{16}$$

Here, we have used the chain rule of mutual information: $\mathcal{I}(s_{t+1}; \{s_t, z\}) = \mathcal{I}(s_{t+1}; s_t) + \mathcal{I}(s_{t+1}; z \mid s_t) \geq \mathcal{I}(s_{t+1}; z \mid s_t)$, resulting from the non-negativity of mutual information. This yield us an information bottleneck style objective where we maximize the mutual information motivated in Section 3, while minimizing $\mathcal{I}(a_t; \{s_t, z\})$. We can show that the minimization of the latter mutual information corresponds to entropy regularization of $\pi(a_t \mid s_t, z)$, as follows:

$$\mathcal{I}(a_t; \{s_t, z\}) = \mathbb{E}_{a_t \sim \pi(a_t|s_t,z), s_t, z \sim p}\left[ \log \frac{\pi(a_t \mid s_t, z)}{\pi(a_t)} \right] \tag{17}$$

$$= \mathbb{E}_{a_t \sim \pi(a_t|s_t,z), s_t, z \sim p}\left[ \log \frac{\pi(a_t \mid s_t, z)}{p(a_t)} \right] - \mathcal{D}_{KL}(\pi(a_t) \mid\mid p(a_t)) \tag{18}$$

$$\leq \mathbb{E}_{a_t \sim \pi(a_t|s_t,z), s_t, z \sim p}\left[ \log \frac{\pi(a_t \mid s_t, z)}{p(a_t)} \right] \tag{19}$$

for some arbitrary distribution $\log p(a_t)$ (for example uniform). Again, we have used the non-negativity of $\mathcal{D}_{KL}$ to get the inequality. We use Equation 19 in Equation 16 to get:

$$R(\pi) \geq \sum_{t=1}^{T-1} \mathcal{I}(s_{t+1}; z \mid s_t) - \beta_t \mathbb{E}_{a_t \sim \pi(a_t|s_t,z), s_t, z \sim p}\left[ \log \pi(a_t \mid s_t, z) \right] \tag{20}$$

where we have ignored $p(a_t)$ as it is a constant with respect to optimization for $\pi$. This motivates the use of entropy regularization. We can follow the arguments in Section 3 to obtain an approximate lower bound for $\mathcal{I}(s_{t+1}; z \mid s_t)$. The above discussion shows how DADS can be motivated from a graphical modelling perspective, while justifying the use of entropy regularization from an information bottleneck perspective. This objective also explicates the temporally extended nature of $z$, and how it corresponds to a sequence of actions producing a predictable sequence of transitions in the environment.

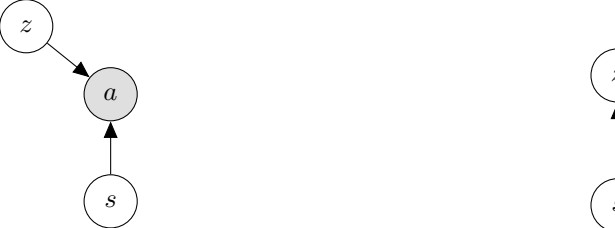

Figure 11: Graphical model for the world $P$ representing the stationary state, action distribution. Shaded nodes represent the distributions we optimize.

Figure 12: Graphical model for the world $N$ using which we is the representation we are interested in.

We can carry out the exercise for the reward function in Eysenbach et al. (2018) (DIAYN) to provide a graphical model interpretation of the objective used in the paper. To conform with objective in the paper, we assume to be sampling to be state-action pairs from skill-conditioned stationary distributions in the world $P$, rather than trajectories. The objective to be maximized is given by:

$$R(\pi) = -\mathcal{I}_P + \mathcal{I}_Q \tag{21}$$

$$= -I(a; \{s, z\}) + I(z; s) \tag{22}$$

$$= \mathbb{E}_\pi[\log \frac{p(z|s)}{p(z)} - \log \frac{\pi(a|s,z)}{\pi(a)}] \tag{23}$$

$$\geq \mathbb{E}_\pi[\log q_\phi(z|s) - \log p(z) - \log \pi(a|s,z)] = R(\pi, q_\phi) \tag{24}$$

where we have used the variational inequalities to replace $p(z|s)$ with $q_\phi(z|s)$ and $\pi(a)$ with a uniform prior over bounded actions $p(a)$ (which is ignored as a constant).

## C  APPROXIMATING THE REWARD FUNCTION

We revisit Equation 4 and the resulting approximate reward function constructed in Equation 6. The maximization objective for policy was:

$$R(\pi \mid q_\phi) = \mathbb{E}_{z,s,s'}\big[\log q_\phi(s' \mid s, z) - \log p(s' \mid s)\big] \tag{25}$$

The computational problem arises from the intractability of $p(s' \mid s) = \int p(s' \mid s, z)p(z \mid s)dz$, where both $p(s' \mid s, z)$ and $p(z \mid s) \propto p(s \mid z)p(z)$ are intractable. Unfortunately, any variational approximation results in an improper lower bound for the objective. To see that:

$$R(\pi \mid q_\phi) = \mathbb{E}_{z,s,s'}\big[\log q_\phi(s' \mid s, z) - \log q(s' \mid s)\big] - \mathcal{D}_{KL}(p(s' \mid s) \mid\mid q(s' \mid s)) \tag{26}$$

$$\leq \mathbb{E}_{z,s,s'}\big[\log q_\phi(s' \mid s, z) - \log q(s' \mid s)\big] \tag{27}$$

where the inequality goes the wrong way for any variational approximation $q(s' \mid s)$. Our approximation can be seen as a special instantiation of $q(s' \mid s) = \int q_\phi(s' \mid s, z)p(z)dz$. This approximation is simple to compute as generating samples from the prior $p(z)$ is inexpensive and effectively requires only a forward pass through $q_\phi$. Reusing $q_\phi$ to approximate $p(s' \mid s)$ makes intuitive sense because we want $q_\phi$ to reasonably approximate $p(s' \mid s, z)$ (which is why we collect large batches of data and take multiple steps of gradient descent for fitting $q_\phi$). While sampling from the prior $p(z)$ is crude, sampling $p(z \mid s)$ can be computationally prohibitive. For a certain class of problems, especially locomotion, sampling from $p(z)$ is a reasonable approximation as well. We want our primitives/skills to be usable from any state, which is especially the case with locomotion. Empirically, we have found our current approximation provides satisfactory results. We also discuss some other potential solutions (and their limitations):

(a) One could potentially use another network $q_\beta(z \mid s)$ to approximate $p(z \mid s)$ by minimizing $\mathbb{E}_{s,z\sim p}\big[D_{KL}(p(z \mid s) \mid\mid q_\beta(z \mid s))\big]$. Note, the resulting approximation would still be an improper lower bound for $R(\pi \mid q_\phi)$. However, sampling from this $q_\beta$ might result in a better approximation than sampling from the prior $p(z)$ for some problems.

(b) We can bypass the computational intractability of $p(s' \mid s)$ by exploiting the variational lower bounds from Agakov (2004). We use the following inequality, used in Hausman et al. (2018):

$$\mathcal{H}(x) \geq \int p(x, z) \log \frac{q(z|x)}{p(x, z)} dx dz \tag{28}$$

where $q$ is a variational approximation to the posterior $p(z|x)$.

$$I(s'; z|s) = -\mathcal{H}(s'|s, z) + \mathcal{H}(s'|s) \tag{29}$$

$$\geq \mathbb{E}_{z,s,s'\sim p}\big[\log q_\phi(s'|s, z)\big] + \mathbb{E}_{z,s,s'\sim p}\big[\log q_\alpha(z|s', s)\big] + \mathcal{H}(s', z|s) \tag{30}$$

$$= \mathbb{E}_{z,s,s'\sim p}\big[\log q_\phi(s'|s, z) + \log q_\alpha(z|s', s)\big] + \mathcal{H}(s', z|s) \tag{31}$$

where we have used the inequality for $\mathcal{H}(s'|s)$ to introduce the variational posterior for skill inference $q_\alpha(z \mid s', s)$ besides the conventional variational lower bound to introduce $q(s' \mid s, z)$. Further decomposing the leftover entropy:

$$\mathcal{H}(s', z|s) = \mathcal{H}(z|s) + \mathcal{H}(s'|s, z)$$

Reusing the variational lower bound for marginal entropy from Agakov (2004), we get:

$$\mathcal{H}(s'|s,z) \geq \mathbb{E}_{s,z}\Big[\int p(s',a|s,z)\log\frac{q(a|s',s,z)}{p(s',a|s,z)}ds'da\Big] \tag{32}$$

$$= -\log c + \mathcal{H}(s',a|s,z) \tag{33}$$

$$= -\log c + \mathcal{H}(s'|s,a,z) + \mathcal{H}(a|s,z) \tag{34}$$

Since, the choice of posterior is upon us, we can choose $q(a|s',s,z) = 1/c$ to induce a uniform distribution for the bounded action space. For $\mathcal{H}(s'|s,a,z)$, notice that the underlying dynamics $p(s'|s,a)$ are independent of $z$, but the actions do depend upon $z$. Therefore, this corresponds to entropy-regularized RL when the dynamics of the system are deterministic. Even for stochastic dynamics, the analogy might be a good approximation , assuming the underlying dynamics are not very entropic. The final objective (making this low-entropy dynamics assumption) can be written as:

$$I(s';z|s) \geq \mathbb{E}_s\mathbb{E}_{p(s',z|s)}[\log q_\phi(s'|s,z) + \log q_\alpha(z|s',s) - \log p(z|s)] + \mathcal{H}(a|s,z) \tag{35}$$

While this does bypass the intractability of $p(s' \mid s)$, it runs into the intractable $p(z \mid s)$, despite deploying significant mathematical machinery and additional assumptions. Any variational approximation for $p(z \mid s)$ would again result in an improper lower bound for $\mathcal{I}(s';z \mid s)$.

(c) One way to a make our approximation $q(s' \mid s)$ to more closely resemble $p(s' \mid s)$ is to change our generative model $p(z,s,s')$. In particular, if we resample $z \sim p(z)$ for every timestep of the rollout from $\pi$, we can indeed write $p(z \mid s) = p(z)$. Note, $p(s' \mid s)$ is still intractable to compute, but marginalizing $q_\phi(s' \mid s,z)$ over $p(z)$ becomes a better approximation of $p(s' \mid s)$. However, this severely dampens the interpretation of our latent space $\mathcal{Z}$ as temporally extended actions (or skills). It becomes better to interpret the latent space $\mathcal{Z}$ as dimensional reduction of action space. Empirically, we found that this significantly throttles the learning, not yielding useful or interpretable skills.

# D    INTERPOLATION IN CONTINUOUS LATENT SPACE

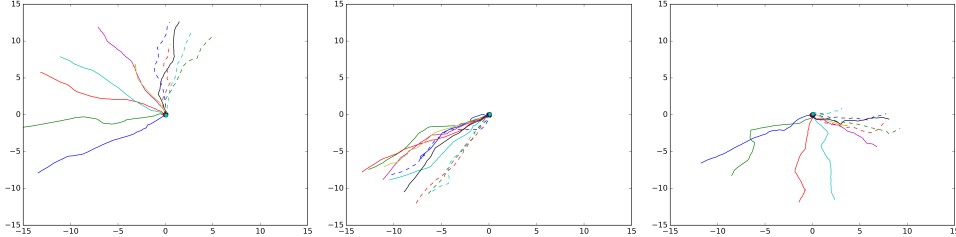

Figure 13: Interpolation in the continuous primitive space learned using DADS on the Ant environment corresponds to interpolation in the trajectory space. (Left) Interpolation from $z = [1.0, 1.0]$ (solid blue) to $z = [-1.0, 1.0]$ (dotted cyan); (Middle) Interpolation from $z = [1.0, 1.0]$ (solid blue) to $z = [-1.0, -1.0]$ (dotted cyan); (Right) Interpolation from $z = [1.0, 1.0]$ (solid blue) to $z = [1.0, -1.0]$ (dotted cyan).

# E    MODEL PREDICTION

From Figure 14, we observe that skill-dynamics can provide robust state-predictions over long planning horizons. When learning skill-dynamics with $x-y$ prior, we observe that the error in prediction rises slower with horizon as compared to the norm of the actual position. This provides strong evidence of cooperation between the primitives and skill-dynamics learned using DADS with $x - y$ prior. As the error-growth for skill-dynamics learned on full-observation space is sub-exponential, similar argument can be made for DADS without $x - y$ prior as well (albeit to a weaker extent).

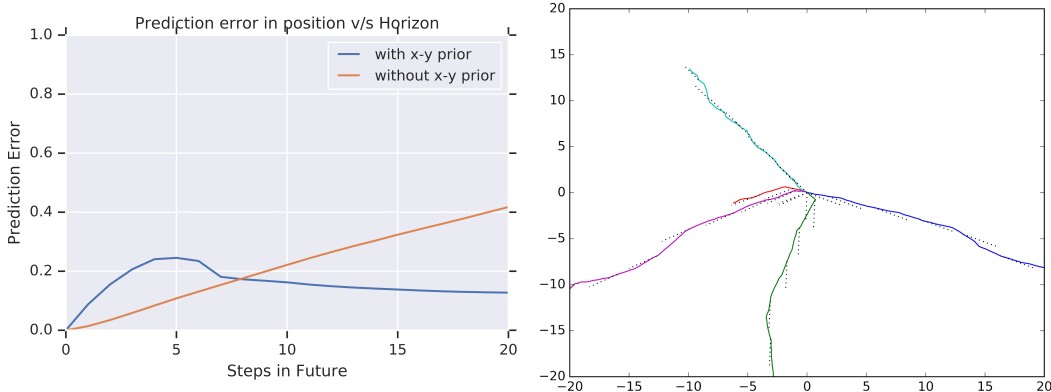

Figure 14: (Left) Prediction error in the Ant's co-ordinates (normalized by the norm of the actual position) for skill-dynamics. (Right) X-Y traces of actual trajectories (colored) compared to trajectories predicted by skill-dynamics (dotted-black) for different skills.

