# OpenReview forum: "Dynamics-Aware Unsupervised Discovery of Skills"
_ICLR.cc/2020/Conference — Accept (Talk)_

### Official Review · AnonReviewer1 · 2019-10-20
**Official Blind Review #1**

**Rating:** 8

**Review:**

This paper proposes a novel approach to learn a continuous set of skills (where a skill is associated with a latent vector and the skill policy network takes that vector as an extra input) by pure unsupervised exploration using as intrinsic reward a proxy for the mutual information  between next states and the skill (given the previous state). These skills can be used in a model-based planning (model-predictive control) with zero 'supervised' training data (for which the rewards are given), but using calls to the reward function to evaluate candidate sequences of skills and actions. The proposed approach is convincingly compared in several ways to both previous model-based approaches and model-free approaches.

This is a very interesting approach, and although I can already think of ways to improve, it seems like an exciting step in the right direction to develop more autonomous and sample efficient learning systems. I suggest to rate this submission as 'accept'.

Regarding the comparison to model-free RL: although it is true that no task-specific training is needed, a possibly large number of calls to the reward functions are needed during planning. It would be good to compare those numbers with the number of rewarded trajectories needed for the model-free approaches.

My main concern with the proposed method is how it would scale when the state-space becomes substantially larger (than the 2 dimensions x and y used in the experiments). The reason I am concerned is that the proposed method uses brutal sampling to search for good trajectories in z-space and action-space. It looks like the curse of dimensionality will quickly make this approach unfeasible. Also, it would be nice to have the learning system discover the important dimensions in which to plan (the x and y in the experiments), rather than having to provide them by hand.

A minor concern is the following: is it possible that the optimization could end up discovering a large number of highly predictable (and diverse) but useless skills?

In the related work section, 1st paragraph, in the list of citations, it might be good to also include the work on maximizing mutual information between representation of the next state and representation of the skill (Thomas et al, arXiv:1802.09484).

The definition of Delta (page 9) is strange: it is said that Delta should be minimized but Delta is defined as proportional to the rewards (which should be maximized). Maybe a sign is missing. Also, why not simply define the rewards as being normalized in the first place, so that the metric IS the accumulated reward rather than this unusual normalized version of it.

**Experience Assessment:**

I have published one or two papers in this area.

**Review Assessment: Checking Correctness Of Derivations And Theory:**

I assessed the sensibility of the derivations and theory.

**Review Assessment: Checking Correctness Of Experiments:**

I carefully checked the experiments.

**Review Assessment: Thoroughness In Paper Reading:**

I read the paper thoroughly.

---

> ### Author Response · Authors · 2019-11-14
> **Response**
>
> (a) The planning algorithm requires ~100,000 calls to the reward function for the given hyperparameters, while the model-free RL will require the same number of reward function calls as the amount of experience required for training (~1M-10M). Note, the number of reward function calls for planning with skill-dynamics is no different from those done in conventional model-based planning algorithms.
>
> (b) We believe the concern about high-dimensional state-spaces is valid. We want to point out that we run DADS on Humanoid on the full state-space (47 dimensional state-space), and we were able to discover different gaits and some x-y coverage as well which could be used for goal navigation (limitedly). We believe in high dimensional state-spaces, DADS will continue to discover skills, but the space of skills discovered might not necessarily be useful for the intended set of downstream tasks, as pointed out by the reviewer. However, we believe this problem can be addressed by automatically learning state-representations which encourage skill-discovery relevant for the downstream tasks. We hope that the future work will look into this line of enquiry.
>
> (c) Thanks for pointing out the reference! We have added it to the related work section and updated the manuscript to reflect the changes.
>
> (d) We thank the reviewer for pointing out the missing negative sign in the metric.  The metric was introduced separately from the reward function so that the navigation to different goals (of differing distances from the origin) can be compared more fairly. However, the reviewer is right to note that the metric itself can be used as the reward function. We have changed the manuscript to clear the confusion.

---

### Official Review · AnonReviewer3 · 2019-10-27
**Official Blind Review #3**

**Rating:** 8

**Review:**

The authors try to incorporate intermediate-level skills into model-based RL, which is an essential problem in the RL field. The algorithm works well even in the case of high-dimensional state and action spaces. Their contributions are in four aspects:
(1)they propose an unsupervised RL framework for learning intermediate-level representations, i.e. skills, based on maximizing the mutual information between the future state and current skill given the current state. This procedure is well-motivated and the mathematics is easy to follow.
(2)they reformulate model predictive control (MPC) in the latent skill space.
(3)their method is compatible with the idea of continuous skill spaces, which seems to give rise to more diverse trajectories and hence offers greater utility.
(4)their method yields low-variance behavior while maintaining enough diversity.

It’s an accept for me. On one hand, using model-free unsupervised RL methods to learn intermediate-level skills is not a new idea. on the other, approaching this problem via mutual information is, as far as I know, new to this field. Although, the novelty of this approach remains undecided, this method seems to work well enough compared to model-based, model-free and hierarchical RL methods. Their analysis from the perspectives of continuous skill space and skill variance also seem to hold.

Nevertheless, the study would benefit from more comparison with other methods using intermediate-level primitives (apart from DIAYN). Moreover it would be interesting to show this method works in scenarios apart from locomotion. I wonder how well the approximation p(z|s) \approx p(z) works in non-locomotion tasks. Otherwise, the authors should mention this method is somewhat limited to locomotion tasks in the main text.

Others:
Typo:
1.Page 3: “maximally informative about about …” remove the redundant “about”;
2.Page 8, first line in section 6.3 “is to be enable use of planning algorithms…” may be changed to “is to take advantage of planning algorithms”.

**Experience Assessment:**

I do not know much about this area.

**Review Assessment: Checking Correctness Of Derivations And Theory:**

I assessed the sensibility of the derivations and theory.

**Review Assessment: Checking Correctness Of Experiments:**

I assessed the sensibility of the experiments.

**Review Assessment: Thoroughness In Paper Reading:**

I made a quick assessment of this paper.

---

> ### Author Response · Authors · 2019-11-14
> **Response**
>
> (a) While there are other methods using intermediate level primitives, prior work in unsupervised skill/option discovery has primarily been demonstrated in discrete control. DIAYN [2] (which matches our assumptions of unsupervised learning + continuous control) was shown to perform better than prior work, for example Variational Information Maximizing Exploration (VIME) [3] and Variational Intrinsic Control [1].
>
> (b) We do hope to demonstrate the utility of this method in manipulation domains. The reviewer is right to note that this may require overcoming the approximation of p(z|s) \approx p(z) for some class of problems. Effectively dealing with this approximation remains a relevant and open question, which we hope will be addressed by future work.
>
> (c) Thanks for pointing out the typos, we have updated the manuscript to reflect the changes.
>
> [1] Gregor, Karol, Danilo Jimenez Rezende, and Daan Wierstra. "Variational intrinsic control." arXiv preprint arXiv:1611.07507(2016).
> [2] Eysenbach, B., Gupta, A., Ibarz, J., & Levine, S. (2018). Diversity is all you need: Learning skills without a reward function. arXiv preprint arXiv:1802.06070.
> [3] Houthooft, Rein, et al. "Vime: Variational information maximizing exploration." Advances in Neural Information Processing Systems. 2016.

---

### Official Review · AnonReviewer2 · 2019-10-28
**Official Blind Review #2**

**Rating:** 8

**Review:**

This paper introduces an unsupervised learning algorithm Dynamics-Aware Discovery of Skills (DADS) for learning low-level “skills” that can be leveraged for model-predictive control. The skills are learned by maximizing the mutual information between the next state s’ and the current skill z conditioned on the current state s. Maximizing this objective corresponds to maximizing the diversity of transitions produced in the environment, while making the skill z be informative about the next state s’. The idea is that using this objective leads to learning a diverse set of skills that are predictive of the environment. The skills z correspond to a set of action sequences, which are represented by a distribution \pi(a|s,z). Because the above objective is intractable to compute (because it relies on the true dynamics p(s’|s,a)), it is variationally lower bounded using the approximate dynamics q_{\phi}(s’|s,z), which represents the transition dynamics when using a certain skill and this variational lower bound is optimized to produce the optimal q_{\phi}(s’|s,z) and \pi(a|s,z).

In the second phase, model predictive control is used to do planning for a new test environment where we have access to the reward function. This corresponds to simulating multiple trajectories using the learned transition dynamics and skill function, computing the reward of each trajectory according to the reward function, executing the first action of the most optimal trajectory and repeating. It is mentioned that planning is done in the latent skill space, which enables easier longer-horizon planning.

Experiments are performed to show that: (1) the learned skills exhibit low-variance behavior (which means that the skills have predictable behavior when used for model predictive control); (2) Model predictive control performs favorably compared to other relevant baselines.

Overall, I feel this is a very well-motivated and interesting submission with very thorough experiments.

**Experience Assessment:**

I do not know much about this area.

**Review Assessment: Checking Correctness Of Derivations And Theory:**

I assessed the sensibility of the derivations and theory.

**Review Assessment: Checking Correctness Of Experiments:**

I assessed the sensibility of the experiments.

**Review Assessment: Thoroughness In Paper Reading:**

I read the paper at least twice and used my best judgement in assessing the paper.

---

> ### Author Response · Authors · 2019-11-14
> **Response**
>
> We greatly appreciate your assessment and thank you for a positive response!

---

### Decision · Program_Chairs · 2019-12-19

**Decision:**

Accept (Talk)

**Comment:**

This is a very interesting paper on unsupervised skill learning based on the predictability of skill effects, with the incorporation of these ideas into model-based RL.

This is a clear accept, based on the clarity of the ideas presented and the writing, as well as the thorough and convincing experiments.